# Comparative Evaluation of Influence of Nd:YAG Laser (1064 nm) and 980 nm Diode Laser on Enamel around Orthodontic Brackets: An In Vitro Study

**DOI:** 10.3390/medicina58050633

**Published:** 2022-05-01

**Authors:** Daliana-Emanuela Mocuta(Bojoga), Oana Grad(Buriac), Marius Mateas, Ruxandra Luca, Darinca Carmen Todea

**Affiliations:** 1Department of Oral Rehabilitation and Dental Emergencies, Faculty of Dental Medicine, Victor Babeș University of Medicine and Pharmacy, 9 Revolutiei 1989 Avenue, 300070 Timisoara, Romania; luca.ruxandra@umft.ro (R.L.); todea.darinca@umft.ro (D.C.T.); 2Interdisciplinary Research Center for Dental Medical Research, Lasers and Innovative Technologies, 9 Revolutiei 1989 Avenue, 300070 Timisoara, Romania; 3Faculty of Industrial Chemistry and Environmental Engineering, Research Institute for Renewable Energy, Politehnica University of Timisoara, Piata Victoriei, 300006 Timisoara, Romania; oana.grad@upt.ro; 4Mechatronics Department, Polytechnic University of Timisoara, 1 Mihai Viteazu Avenue, 300006 Timisoara, Romania

**Keywords:** 980 nm diode laser, Nd:YAG, energy-dispersive X-ray, micro-hardness, laser, scanning electron microscopy

## Abstract

(1) Background: The prevention of demineralizing lesions at the enamel structure level continues to represent a challenge in daily dental practice. When bacteria influence the pH level, this will decrease below the threshold for remineralization and the dissociation of hydroxyapatite will occur with a high percentage of phosphate and calcium loss. These elements continue to be studied by many authors in order to obtain a working protocol that will lead to their stabilization at the level of the enamel structure, thus preventing the demineralization process. The aim of this study is to evaluate and compare the influence of two types of laser wavelengths on the surface morphology and mineral components of the enamel through an examination with scanning electron microscopy (SEM) and energy-dispersive X-ray spectrometry (EDX). (2) Methods: Thirty permanent human incisors extracted for periodontal reasons from patients aged 25–40 years old were selected for this study. Metallic brackets (SS Standard 022 Slot, OC Orthodontics, McMinnville, OR, 97128, USA) were bonded onto each tooth. The buccal surface was randomly assigned three sections: Section A—negative control (no treatment), section B—treated with 980 nm Gallium–Aluminum–Arsenide diode laser (a 300 µm optic fiber was used with 0.8 W output power, energy density of 5.33 J/mm^2^, in continuous mode, for 30 s, oriented perpendicularly to the enamel surface in contact mode) (KaVo GENTLEray 980 Diode Laser, Kaltenbach & Voigt GmbH, Biberach, Germany), and section C—treated with Nd:YAG laser (a 300 µm fiber was carried out at a 1 mm distance from the enamel surface with 0.75 W power, 75 mJ pulse energy, pulse repetition rate of 10 Hz, 5 J/mm^2^ fluency, average exposure time of 30 s, and water cooling assisted) (LIGHTWALKER AT S, M021-5AF/1 S, Fotona d.o.o, Ljubljana, Slovenia). The elements evaluated in this study were calcium (Ca), phosphate (P), oxygen (O), and carbon (C). A one-way analysis of variance, paired t-tests, and independent t-tests were carried out to evaluate the results using the SPSS 19 IBM Statistical package software for Microsoft. (3) Results: The evaluation of the data indicated that both wavelengths produced an increase in Ca wt% (for diode laser the mean of Ca wt% before irradiation was 21.06, while that after treatment reached 28.24; and for Nd:YAG laser, the mean of Ca wt% before irradiation was 21.31, while that after treatment reached 33.88); as well, the 980 nm diode laser decreased P wt% (from 17.20 before irradiation to 16.92 after irradiation) and the Nd:YAG laser increased P wt% (from 17.46 before irradiation to 18.28 after irradiation). These results showed a statistically significant difference at the *p* < 0.05 level. (4) Conclusions: It can be concluded that the best improvement of enamel chemical composition was obtained with Nd:YAG irradiation.

## 1. Introduction

The prevention of demineralizing lesions at the enamel structure level continues to represent a challenge in daily dental practice. The clinical aspect of white spot lesions represents the substrate underlying the appearance of carious lesions in the presence of an acidic environment. Demineralization phenomena produce an advanced impairment of the enamel mineral composition, which is often irreversible, thereby invoking the need to follow a restorative procedure for the affected tooth. Multi-factorial causes (e.g., acidic pH, high-carbohydrate diet, excessive consumption of carbonated beverages, local conditions represented by elements that retain the bacterial plaque and make it difficult to clean in those areas-orthodontic brackets, immobilization rails, and patients with motor deficits) that underlie the demineralization phenomenon can only be offset by applying specific preventive methods associated with a good oral hygiene education.

The fixed orthodontic appliances act as a food stagnation area and increase the potential for the development of dental plaque. When the bacteria work on fermentable carbohydrates, it results in a decrease of the pH level below the threshold for remineralization, and then the carious decalcification occurs [1].

At present, the methods used to increase the micro-hardness of the enamel and reduce its solubility include topical applications of fluoride, a reduction in cariogenic foods and refined carbohydrates, daily removal of bacterial plaque, and the use of antimicrobial agents [1,2].

According to specialized studies on the incidence of dental carious lesions, it can be said that populations with a poor socioeconomic status present a high risk of having caries as well as large families with members of varying degrees of dental malocclusion who have a high incidence of dental caries. In these cases, the prevalence of carious lesions increases from 10.8% to 50%, according to Borzabadi-Farahani et al. [3]. In these cases, the application of combined preventive therapies can lead to better control and, consequently, to the prevention of carious lesions [2,4]. The demineralization phenomenon is a reversible one; initially, it takes place in the presence of a pH lower than 5.5, leading to the dissociation of hydroxyapatite (HA), which represents 95% of the mineral part of enamel. In the natural presence of calcium and phosphorus ions from the saliva, a change in pH occurs, which favors the balancing of demineralization. If this phenomenon is not effective or it is affected, the progressive loss of minerals continues, leading to the demineralization of the enamel layer [5,6].

As is well known, orthodontics continues to pose a major challenge for dentists in terms of improving and reducing the negative effects that the retentive elements of the brackets represent. Thus, it has been emphasized that the most prone areas for the accumulation of bacterial plaque at the enamel surface are adjacent to the bracket base. These negative effects are even more obvious after bracket debonding, when the aesthetic aspect of the vestibular surface is altered, thus manifesting the white spot lesions [7,8,9,10].

At present, the use of laser radiation is considered an additional preventative method for the carious lesions at the enamel structure level. This is possible due to the fact that some wavelengths—if used with the correct and appropriate working parameter settings—can produce changes in the chemical microstructure of the enamel, leading to a reduction in its solubility [11]. Recent research has shown positive results with the use of infrared laser radiation on enamel, both as a single therapy method and in combination with various remineralizing agents [12]. Moreover, laser-assisted treatments have an advantage that most patients accept, namely the lack of vibrations and reduced working hours, which will greatly influence the decision to accept those treatments [13].

Studies conducted on laser radiation by several authors have led to the encouragement of its use in prevention. Wavelengths that have achieved a marked acid resistance of the enamel are 9.6 μm and 10.6 μm (CO_2_) [14], 2.94 μm (Er:YAG) [15], and 2.79 μm (Er:YSGG) [16,17]. Nd:YAG (1.064 µm) and Argon [18] lasers have also been shown to be successful, with promising results in reducing enamel solubility. High-power lasers (e.g., CO_2_) have been shown to increase enamel hardness and reduce its solubility [19,20,21]. However, these lasers should be used with caution, adopting a low energy level to maintain the integrity of the enamel and minimizing the risk of microporous areas on its surface [22].

Moreover, low-power lasers have been studied by a few authors in relation to the management and prevention of demineralization processes in dental structures and positive results for enamel microhardness increases have been achieved [23,24]. Reports on the use of the 810–830, 940, 980, and 1064 nm wavelengths are more frequent in the literature. Diode lasers are considered to be ideal for daily practice treatments of orthodontics, such as soft tissue procedures leading to sufficient haemostasis and precise incision margins [25]. Also, semiconductor diode lasers have been studied as an alternative to conventional procedures as a way of improving the hardness of the enamel, reducing the solubility of the mineral component, and inhibiting enamel demineralization, which has led to positive results that are of real use in the prevention of carious lesions [26]. Diode lasers are much more accepted in clinical practice due to the small size of the equipment, the low cost, and the ease of handling the handpiece, which facilitates access to the oral cavity [27,28].

In recent years, the interest of scientific research on orthodontic fixed therapy prevention has focused on chemical changes at the enamel structure level during its interaction with the laser radiation [29]. Special care must be taken in laser use in order to minimize the negative effects to the surrounding tissues [22]. The evaluation of inorganic changes in the composition of enamel can be performed using spectroscopic techniques, including wavelength-dispersive X-ray spectroscopy (WDXS or WDS), X-ray photoelectron spectroscopy (XPS), Energy Dispersive X-ray Spectroscopy (EDS, also called EDX or XEDS), and Fourier-transform Raman and infrared spectroscopy (FT-Raman and FTIR) [30,31].

In the context of all these shortcomings, in terms of prevention and its role in fixed orthodontics, the aim of this study is to evaluate and compare the effects of two laser wavelengths on the enamel structure adjacent to orthodontic brackets, as well as to analyze the composition of chemical elements and enamel surface structure. This research finds its applicability in seeing and observing the quality of a laser-assisted preventive treatment in order to minimize the effects of bacterial plaques that are difficult to remove from around the brackets, as well as its impact on the enamel in terms of changing its chemical structure.

For the present study, the following null hypotheses were considered:There will be no changes to the enamel surface before and after the experimental procedure.There will be no difference in the percentage of chemical elements in the assessed areas before and after laser irradiation.The two wavelengths will not differently influence the chemical composition of the enamel.

## 2. Materials and Methods

The study protocol was approved by the Research Ethics Committee of “Victor Babes” University of Medicine and Pharmacy in Timisoara, Romania (approval number 57 from 6 December 2021). This study was conducted at the Faculty of Dentistry in collaboration with the Research Institute for Renewable Energy, Timisoara. All subjects enrolled in this study were informed regarding our research protocol and signed an informed consent form regarding the use of their data and samples for scientific purposes.

The experimental part included two phases: a clinical one with tooth selection and an “in vitro” one with tooth preparation, experimental procedures, and, finally, structural morphology and chemical element analysis of the enamel.

### 2.1. Tooth Selection and Preparation Phase

Thirty permanent human incisors extracted for periodontal reasons from patients aged 25–40 years old were used for this study. The exclusion criteria were: teeth presenting any defects, microcracks, erosions, caries lesions, restorations, and visible defects of buccal surface. The inclusion criterion was: intact teeth extracted from 25–40 years old adult patients with periodontal disease. The selected teeth were stored in saline solution at 4 °C after extraction, which was changed every 24 h until the experiment was performed.

After tooth selection, cleaning with an ultrasonic scaler (EMS miniPiezon, SA CH-1260 Nyon, Switzerland) and brushing with fluoride-free paste (Clean Polish, KerrHawe SA, 6934 Bioggio, Switzerland) were performed. On the buccal surface of each tooth, we bonded metallic brackets (SS Standard 022 Slot, OC Orthodontics, McMinnville, OR, 97128, USA), following the manufacturer’s instructions for each product. The demineralization time of the buccal surface was 30 s, using 37% orthophosphoric acid (Ormo Etching Solution, Ormco, Brea, CA 92821, USA) and followed by 20 s of washing and drying. A primer (3M, Transbond XT, Unitek, Monrovia, CA 91016, USA) was applied and the metallic brackets were bonded using Transbond Plus Color Change (3M, Unitek, Monrovia, CA 91016, USA). Excess resin from the bracket base was removed using a dental probe. The brackets were polymerized for 10 s on each side using an LED lamp (420–480 nm, 1500 mV/cm^2^; Rainbow Curing light, Henan, China).

The buccal surface of the specimens was randomly assigned to three section-groups (areas), as shown in Figure 1: Section A—negative control group (no treatment), Section B—Diode Group—treated with a 980 nm diode laser (KaVo GENTLEray 980 Diode Laser, Kaltenbach & Voigt GmbH, Biberach, Germany), and Section C—Nd:YAG group—treated with a Nd:YAG laser (LIGHTWALKER AT S, M021-5AF/1 S, Fotona d.o.o, Ljubljana, Slovenia). To irradiate and evaluate the precise area selected for the study, we considered windows situated near the bracket limits in the mesial, distal, and apical directions. The dimensions of the areas were set with a 3 mm height and 1.5 mm width for Sections B and C, and a 3 mm width and 1.5 mm height for Section A. This was made possible by marking the extreme points of the selected working regions with a marker and taking into account the marginal dimensions of the bracket.

### 2.2. Experimental Procedures

The 980 nm diode laser irradiation conditions were as follows. The diode laser was applied with a 0.8 W output power, an energy density of 5.33 J/mm^2^, and in continuous mode to Section B for 30 s. A 300 µm optic fiber conductor, as a transmission element, was attached and oriented perpendicularly to the enamel surface. The optic fiber tip was used in contact mode, and irradiation was performed by hand by the same operator who screened the enamel surface in a uniform motion in order to cover the entire selected area.The Nd:YAG laser irradiation conditions were as follows. The laser was used with a fixed wavelength of 1064 nm, a power of 0.75 W, a 75 mJ pulse energy, a pulse repetition rate of 10 Hz, an energy density of 5 J/mm^2^, an average exposure time of 30 s, and with assisted water cooling. The irradiation mode was carried out at a 1 mm distance from the enamel surface with a 300 µm fiber in a sweeping movement that scanned once in a horizontal direction in order to cover the entire tested area and was done by the same operator to avoid human errors.

### 2.3. Scanning Electron Microscopy (SEM) and Energy Dispersive X-ray Spectroscopy (EDX) Analysis

To analyze the enamel surface morphology and chemical composition at delimited sections near the brackets, the samples were characterized by scanning electron microscopy and elemental analysis X-ray energy dispersive spectroscopy EDX by using an FEI Quanta FEG 250 instrument (SEM-EDX, Type Quanta 250 FEG, Model No 1027641, FEI company, Brno, 62700, Czech Republic) at the Research Institute for Renewable Energy, Politechnica University of Timisoara, Romania.

The surface morphology was analyzed in the negative control group—Section A, and before and after the experimental procedures for Sections B and C at ×1000 magnification (Figure 2).

The elemental analysis was performed for all three sections for each tooth; however, for the experimental groups, the evaluation was done before and after laser irradiation. The EDX system operated as an integrated part of the SEM Quanta FEG 250 instrument. This is an X-ray technique used to identify the elemental composition of materials. The data generated by the EDX analysis consist of spectra showing peaks corresponding to the elements making up the true composition of the sample being analyzed.

With the X-ray microanalyzer EDX, the quantitative number of chemical elements from the studied sections were recorded. The surfaces were examined by EDX, which was operated at 15 keV (kiloelectronvolts) using high vacuum mode, a working distance (WD) of 10 mm, an aperture Ø of 50µm, a measurement time of 45 microseconds, and an image resolution of 1024 × 800 pixels, at ×4000 magnification. In each study area, a centrally located region of interest was defined and one field (49 × 38 µm at a magnification of ×4000) was defined for elemental analysis. The elements evaluated in this study were calcium (Ca), phosphorus (P), oxygen (O), and carbon (C), and their contents are expressed in weight percentage (wt%) (Figure 2). Due to the storage conditions, sodium (Na) and chlorine (Cl) elements were detected during the elementary EDX analysis, which were not considered significant for the enamel structure during the evaluation data phase, and thus were not taken into consideration as they have no influence on the enamel microhardness. The weight percentages of Ca and P are considered to be very important in influencing the microhardness of the enamel, as many studies have shown previously [32,33,34].

### 2.4. Statistical Analysis

In order to compare the overall differences in the chemical element composition before and after the experimental procedure, a statistical analysis of the obtained data was performed using one-way analysis of variance (ANOVA). A paired t-test was carried out to check whether a significant difference was observed between the conditions before and after the experimental procedure and to establish if a main effect existed in the recorded data. Moreover, we analyzed whether there was a statistically significant difference between the levels of Ca (wt%) and P (wt%) recorded before and after laser irradiation. For this reason, an independent t-test was applied to determine the difference between the effects of the two considered lasers. The statistical analysis was performed using the SPSS 19 IBM Statistical package software for Microsoft (SPSS IBM, New York, NY, USA). The statistical significance level was set at *p* < 0.05 for the entire assessment and the statistical confidence level was 95%.

## 3. Results

The morphology of the enamel surface was captured before and after laser irradiation with different wavelengths applied to each sample. Figure 3 and Figure 4 show the surface morphology of the control enamel area, as well as the surface topography recorded before and after 980 nm diode laser irradiation at ×1000 magnification. The SEM and EDX records shown in Figure 3 and Figure 4 are from two randomly evaluated teeth. The surface appeared smooth and uniform for both the control group and Section B before laser irradiation. After diode laser irradiation, the appearance of the enamel changed to a less uniform surface, with some unevenness.

The X-ray energy dispersive spectroscopy elemental analysis of enamel structure composition was carried out for control Section (A) and both of Sections (B) and (C), where the experimental procedures were applied. The examination of different enamel areas identified the elements Ca, P, C, and O, as well as other elements (i.e., Cl and Na), according to their weight percentage. Table 1 presents the mean weight percentages of Ca, P, C, and O and their concentrations for the control section, as well as their initial values (i.e., assessed before irradiation) and their values after the experimental procedure involving 980 nm diode laser irradiation.

The surface morphology of Section C before and after Nd:YAG laser irradiation is shown in Figure 5 and Figure 6. Again, the SEM images and EDX results are from two randomly assessed teeth. Initially, the enamel surface appears uneven, presenting tiny holes and roughness with indentations before laser irradiation. After Nd:YAG laser irradiation, the appearance of the enamel is characterized by a smooth, relatively uniform, and crack-free surface with small irregularities. Table 2 shows the mean of the weight percentages of Ca, P, C, and O identified before and after laser irradiation with Nd:YAG.

Table 3 presents the paired means of the identified elements by weight percentage before and after the experimental procedure, which were recorded for Sections B and C. It can be seen that an increase in Ca was obtained after both irradiation treatments (i.e., the 980 nm diode laser and the Nd:YAG laser). For P, an increase was obtained only after Nd:YAG irradiation. After diode irradiation, the wt% of P decreased for most samples. Furthermore, the wt% of O, C, Na, and Cl were lower after laser irradiation compared to that in the initial evaluation.

Table 4 shows the percentage values of the mean differences in the contents of Ca, P, O, C, and other elements (Na, Cl) between the different experimental phases, which were recorded following EDX analysis. For this, a paired t-test was applied to all the evaluated chemicals elements, thereby comparing the mean of the initial concentration with that obtained after the laser irradiation. Statistically significant differences were observed for all chemical elements in the evaluation except for P, for which no statistically significant differences were recorded after application of the 980 nm diode laser. Furthermore, no statistically significant differences were observed for the mean of Na wt% and Cl wt% after both irradiation treatments.

Figure 7 and Figure 8 show graphical representations of the mean difference in weight percentage (wt%) of the element’s calcium and phosphorus, respectively, which were recorded during the experimental procedure (i.e., before and after laser irradiation). The difference between the initial and final measurement of Ca was higher for the Nd:YAG section (C) than for the 980 nm diode laser section (B). Nd:YAG irradiation produced an increase in P wt%, while, for most samples, the diode laser led to the maintenance of or even a reduction in phosphorus levels. This suggests that the main effect of Nd:YAG irradiation treatment is increasing calcium and phosphorus concentrations in the enamel structure, thus making it more resistant to the solubility process.

Evaluating the results from a statistical point of view, a difference was observed regarding the influence of the 980 nm diode laser and the Nd:YAG laser on the calcium and phosphorus levels. As a result, we hypothesized that the Nd:YAG laser increases the wt% of Ca more than irradiation with a 980 nm diode laser when compared to the initial value of Ca wt% in the target area. Moreover, the phosphorus levels after irradiation with a 980 nm diode laser was reduced compared to the initial value, while it was increased after irradiation with the Nd:YAG laser. This hypothesis was verified through an independent t-test conducted using the IBM SPSS software (see Table 5 and Table 6).

According to the results of the independent t-test, the hypothesis that there was a difference between the 980 nm diode laser (used for Section B) and the Nd:YAG laser (used for Section C), regarding the modification of Ca and P was accepted. Due to this, the null hypotheses were rejected. Both wavelengths produced a change in the weight percentage of Ca and P compared to the initial composition. The results refer to a statistically significant difference in these results at the *p* < 0.05 level.

## 4. Discussion

The development of the initial demineralization lesions on the enamel surface is the most important negative effect produced by fixed orthodontic appliances, which are known as white spot lesions [33]. In the related literature, studies have estimated these lesions as having an incidence between 50% and 70% [35], while Sundararaj et al. [36] have situated the incidence of new carious lesions formed during orthodontic treatment at 45.8% and their prevalence at 68.4%. As is well known, prevention should be considered an essential objective for oral–dental health status.

The most effective studied method used for improving the resistance of enamel to the demineralization process is fluoride therapy [7,8,9,14,21,23,28]. It has been consistently demonstrated that high-powered lasers, under certain conditions, influence the solubility of the enamel and, thus, reduce its demineralization rate. This is made possible by altering the enamel crystalline structure. Also, low-level lasers (used alone or with topical fluoride) can lead to an increase in enamel resistance under an acid environment, as has been reported in the previous literature [23,32,34,37,38,39]. In other studies, nanoparticles were used in order to prevent bacterial adhesion and reduce enamel demineralization. The results seem promising, but future research in this area is needed [40].

Since their introduction in 1962, the diode laser family has grown considerably and diode lasers with wavelengths in the range of 445–2200 nm have been used for the treatment of various medical conditions. The diode lasers can be used in a continuous-wave mode and are the ideal choice for use in orthodontic set-up because of the smaller size (“footprint”) of the laser device and relatively lower cost involved [25].

In the context of the shortcomings of fixed orthodontic treatments, in terms of the effects on enamel as well as the methods to prevent early demineralization (assisted or not by laser therapy), the aim of this study was to evaluate and compare the effects of two wavelengths on the enamel surface morphology and chemical composition of enamel adjacent to orthodontic brackets. For this study, we selected the 980 nm and 1064 nm wavelengths from considering reports from the literature with positive results in terms of increasing enamel resistance [18,32,33,37,38,39,40,41,42,43,44,45,46,47,48,49]. The assessment of enamel surface morphology before and after the experimental procedure and chemical composition analysis of the tested areas near the brackets were carried out through scanning electron microscopy and elemental analysis X-ray energy dispersive spectroscopy (EDX) using an FEI Quanta FEG 250 instrument (SEM-EDX, Type Quanta 250 FEG, Model No 1027641, FEI company, Brno, 62700, Czech Republic).

In the present study, the evaluated chemical elements were calcium, phosphorus, oxygen, and carbon, which were recorded in terms of their weight percentages. Before treatments, the mean of Ca wt% for Section A (control) was 21.46, for Section B (980 nm laser) it was 21.06, and for Section C (1064 nm laser) it was 21.31. By observing this aspect, we can conclude that the concentration of calcium varies in a very insignificant manner on the enamel buccal surface. The reason for this could be due to the enamel’s thickness, the age of the patient at which the extraction was performed, their eating habits, and their level of oral hygiene [32,50]. Furthermore, the concentration of P wt% was almost the same for the control and the other two evaluated areas before laser irradiation, as can be seen in Table 1 and Table 2. Minor differences were also observed for the elements C, O, Na, and Cl in the examined areas; however, due to the storage conditions of the teeth, Na and Cl were not considered as positively or negatively influencing the solubility of the enamel [4,51,52].

Following the experimental procedure using both wavelengths, there were changes in the contents of all elements studied. A statistical analysis of the recorded data indicated some significant differences when comparing the composition of the enamel after irradiation. The 980 nm diode laser group showed an increase in the mean for Ca wt%, from 21.06 to 28.24, and a loss of P wt%, from 17.20 to 16.92. Due to the differences in the phosphorus weight percentage recorded before and after laser irradiation, it can be stated that there was no statistically significant difference in the weight percentage of this element (see Table 4, pair 3). Furthermore, the differences between mean O wt% and C wt% were statistically significant, with decreases recorded in both. Our results assessing the 980 nm diode laser were in agreement with previous studies that have obtained enamel that is less susceptible to acid demineralization due to calcium increases, less-modified phosphorus concentrations, and decreases in carbon and oxygen contents [33,37,53].

The areas irradiated with the Nd:YAG laser showed a higher percentage of calcium compared to those treated with the diode laser. The mean of Ca wt% before irradiation was 21.31, while after treatment it reached 33.88. Moreover, the content of phosphorus increased from 17.46 to 18.28 after irradiation. Both the O wt% and the C wt% decreased. The statistical evaluation showed statistically significant differences in all four element contents before and after Nd:YAG laser irradiation. The results obtained for Section C (Nd:YAG) were in accordance with the results of other articles in the literature, which have obtained an increase in the hardness of the enamel after laser irradiation due to modification of the chemical elements in its structure [18,32,43,45].

In another study, the authors evaluated the effect of an Nd:YAG laser on the enamel layer and they observed an inhibition of dental caries development, which could promote both remineralization and acid resistance on the enamel surface by changing its crystallization [54]. Regarding their conclusions and compared to our results recorded after Nd:YAG irradiation, we can say that the wavelength of 1064 nm could change the crystal structure of the enamel and make it more resistant to acid attack.

In relation to the surface morphology assessed before and after treatment with the 980 nm diode laser, it can be stated that no major effects occurred on the tested enamel area, with no fissures or cracks having been provoked. The SEM results of our study are in agreement with those obtained by Umana et al. [37] and Nandkumar et al. [55], who have shown that 980 nm and 810 nm diode lasers at 0.8 W and 1 W, respectively, do not cause damage to the enamel and dentin surface, thereby concluding that diode lasers can provide a non-invasive, pain-free, and almost safe treatment option.

Irradiation of the enamel surface by the Nd:YAG laser led to a smooth, flat, and almost perfectly homogeneous enamel surface with several superficial streaks compared to the area before irradiation. These results are in agreement with Suhaimi et al. [32] and Noorsyazwani et al. [44], who obtained a smooth morphology with small bubbles on the tested area and minimal crack formation after the application of a Nd:YAG laser at 1.5 W with a range of fluences between 80 and 110 J/cm^2^ under a water–air spray. Furthermore, El Mansy et al. [18] and Al-Jedani et al. [41] have observed, through SEM evaluations, smooth areas after irradiation with Nd:YAG at 0.8 W. This aspect is in agreement with our results after Nd:YAG irradiation at 0.75 W. Moreover, the authors clearly stated that the 0.8 W laser obtained the best results regarding the morphology and elemental analyses; meanwhile, the 0.5 W laser did not produce any apparent effect, and the 1.2 W laser produced an ablative effect on the enamel surface [18].

Based on the specific literature [18,32], the effect of laser irradiation on the enamel structure varies. This is most probably due to certain variables, such as power, pulse width, pulse rate, fluence, and time of irradiation. For this reason, the choice of correct parameters for different clinical applications is very important. However, the teeth selected as samples also affect the outcome because the surface morphology and chemical composition of the enamel can vary. As can be seen in the statistical interpretation, the average chemical element contents in the control area (Section A) and the areas examined before laser irradiation were slightly different.

As the literature shows, enamel decalcification represents a significant problem in the orthodontic field. Authors have also studied the argon laser, which has been shown to reduce decalcification during an acidic challenge in vitro. The teeth were examined under polarized light microscopy and the results showed that argon laser irradiation is effective in reducing enamel decalcification during orthodontic treatment [56]. 

Both lasers used in our study increased the concentration of Ca. As for P, this increase was possible only for Section C after Nd:YAG irradiation, while, in Section B—after 980 nm Diode Laser irradiation—a decrease in the wt% mean of P was recorded. We were interested in observing the situation of calcium and phosphorus contents and the effects of the two wavelengths on them because, in the literature, Ca and P are considered the most important elements in the composition of hydroxyapatite [18].

It is well known that increases in Ca wt% and P wt% increase the resistance of enamel to dissolution [7,8,9,11,12,13,14,15,16,17,18,19,20,21,22,23,24,25,26,33,37,39,43,44,45,46,47,48,49,50,51,52,53]; however, the mean of oxygen wt% before and after irradiation for both wavelengths were about the same, while that for carbon was lower after irradiation with both lasers. The content of carbon in teeth can be correlated with various factors, such as enamel maturity, dietary intake, caries susceptibility, and the hypoplastic status of the enamel [32]. Pereira DL et al. [57] have stated that the reduction of carbon can replace the phosphorus or hydroxyl radicals in the biological crystals of apatite in order to form a less-soluble phase. This reduction in carbon content leads to an increase in the resistance of the enamel to the demineralization process.

For this study, we performed an independent t-test to compare the effects of both lasers with respect to the Ca and P contents. Considering the pairwise comparison results obtained in the current work, the difference between Section B and Section C was statistically significant regarding the Ca percentage increase; however, no significant difference was recorded for the P content after diode laser irradiation.

Based on the literature study, calcium hydroxyapatite [Ca_10_(PO_4_)_6_(OH)_2_] is the main mineral component of the tooth structure. Enamel is formed from an inorganic matrix (96%) and organic components such as proteins and lipids, as well as some water (4%). Mature dentine is about 70% mineral, 20% organic matrix, and 10% water by weight. Considering those aspects, adding elements into the crystalline structure can modify the physico-chemical properties, mechanical properties, and the solubility of the hydroxyapatite crystals. For instance, tooth hardness can be improved by adding zinc, while magnesium makes the substrate more porous, and fluoride makes the tooth structure more resistant to acid attack. However, in patients with xerostomia, Sjogren’s syndrome, radiation caries, and older adults who are home-bound due to impaired physical and mental disabilities and have difficulty in carrying out their daily oral hygiene regimen, brushing once daily with a high-fluoride dentifrice would be a more pragmatic approach. The findings may also have implications for the management of white spots in high-risk orthodontic patients [58].

Following the interpretation of the data from a statistical point of view, the null hypotheses established at the beginning of the study were rejected and the research hypotheses were accepted, by which it can be inferred that laser radiation influences the chemical composition of the enamel and that there are differences between the effects of the two different wavelengths on enamel surface morphology.

Regarding the limitations of this study, the following aspects can be highlighted: it is almost impossible to select teeth with identical chemical structures or, at most, which have a similar concentration of chemical elements. The surface morphology also varied from sample to sample. Due to the maturity, density, and calcification of the enamel, all of these factors had an influence on the results of the study. Therefore, a comparative assessment of the surface morphology and composition of the chemical elements in this study was performed on the same areas before and after the application of laser radiation. As a possible suggestion for future research, a much larger number of samples could prove even more clearly the statistical significance regarding the variation in the chemical composition and the increases in enamel microhardness.

## 5. Conclusions

Both laser radiation treatments (i.e., at 980 nm and 1064 nm) considered in this paper were found to affect dental hard tissue in relation to the surface morphology and chemical composition, as demonstrated by the results of this study. It can be concluded that the most important improvement in enamel chemical composition was obtained after Nd:YAG irradiation, which served to increase the acid resistance of the enamel. Considering the lack of side effects, the diode laser and Nd:YAG laser seem to be promising treatment tools in the field of preventive dentistry and clinicians should refer to laser treatment in order to improve the efficiency of orthodontic treatment in routine clinical practice. Further studies are needed to verify the roles of laser parameters in order to determine the optimal conditions for enhancing enamel resistance against solubility processes.

## Figures and Tables

**Figure 1 medicina-58-00633-f001:**
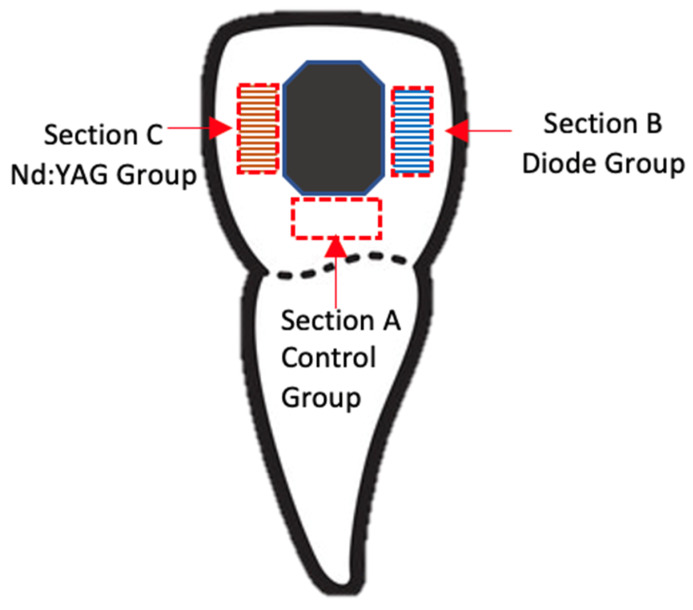
Schematic representation of the tooth with delimitation of control group (Section A) and treatment groups (980 nm Diode Laser—Section B: blue area, 1064 nm Nd:YAG Laser—Section C: orange area). The direction movement of the laser fiber is represented by parallel lines on the coloured areas.

**Figure 2 medicina-58-00633-f002:**
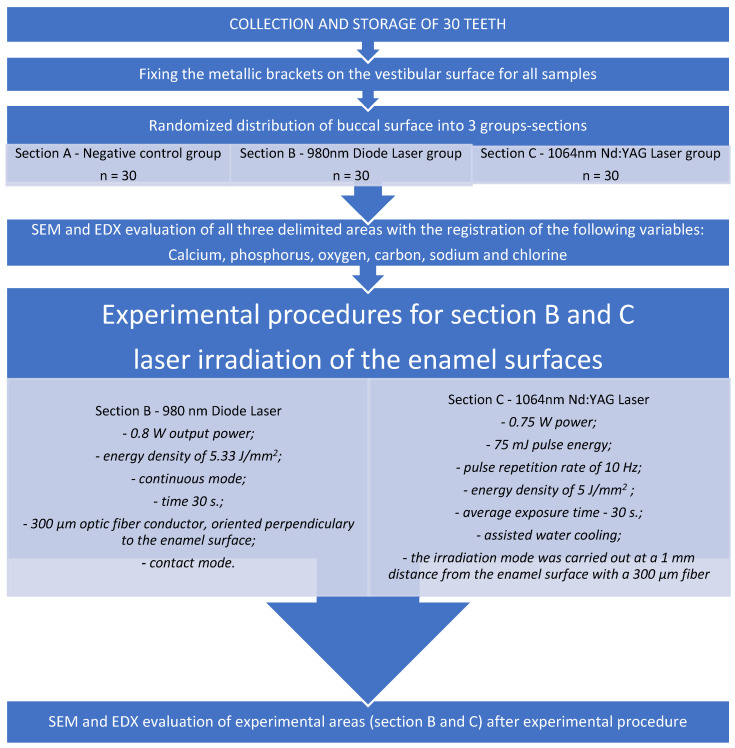
The consort type diagram of the experimental procedure, including the control group (Section A), treatment groups (Section B—980 nm Diode Laser, Section C—1064 nm Nd:YAG laser), and variables measured.

**Figure 3 medicina-58-00633-f003:**
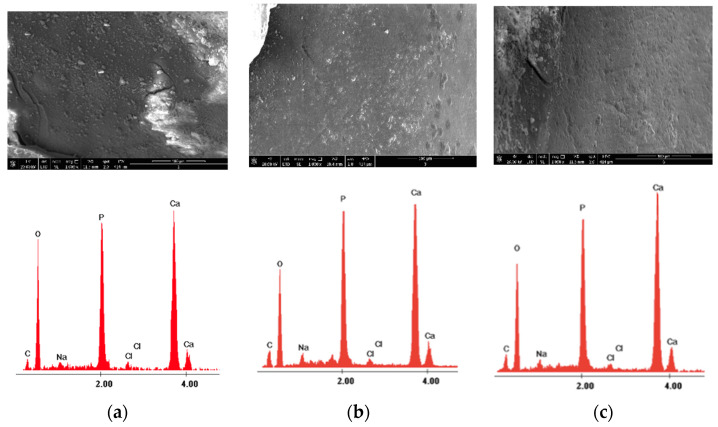
The images recorded by scanning electron microscopy and energy dispersive X-ray analysis images of study groups: (**a**) control; (**b**) before 980 nm diode laser irradiation; and (**c**) after 980 nm diode laser irradiation.

**Figure 4 medicina-58-00633-f004:**
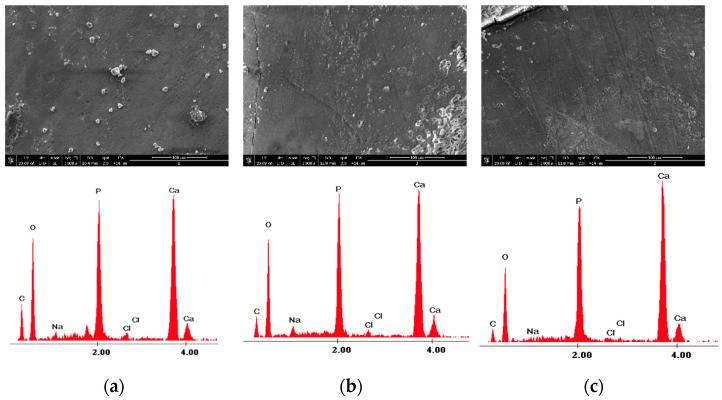
The images recorded by scanning electron microscopy and energy dispersive X-ray analysis images of study groups: (**a**) control; (**b**) before 980 nm diode laser irradiation; and (**c**) after 980 nm diode laser irradiation.

**Figure 5 medicina-58-00633-f005:**
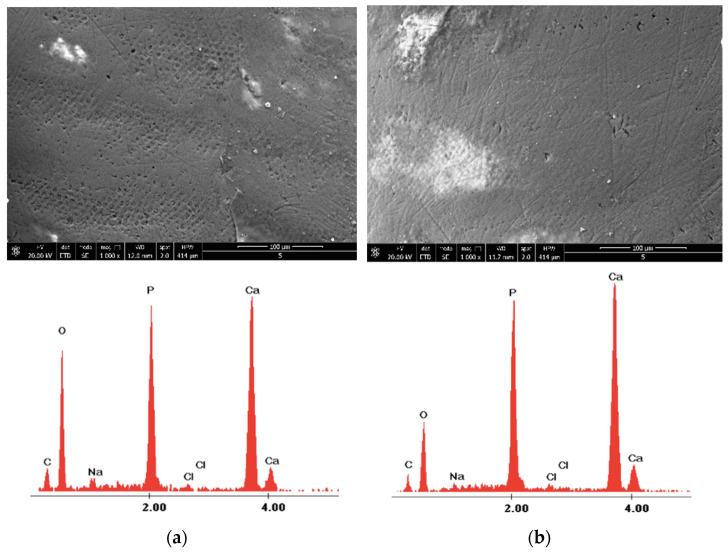
The images recorded by scanning electron microscopy and energy dispersive X-ray analysis for 1064 nm Nd:YAG Laser Group: (**a**) before laser irradiation; and (**b**) after laser irradiation.

**Figure 6 medicina-58-00633-f006:**
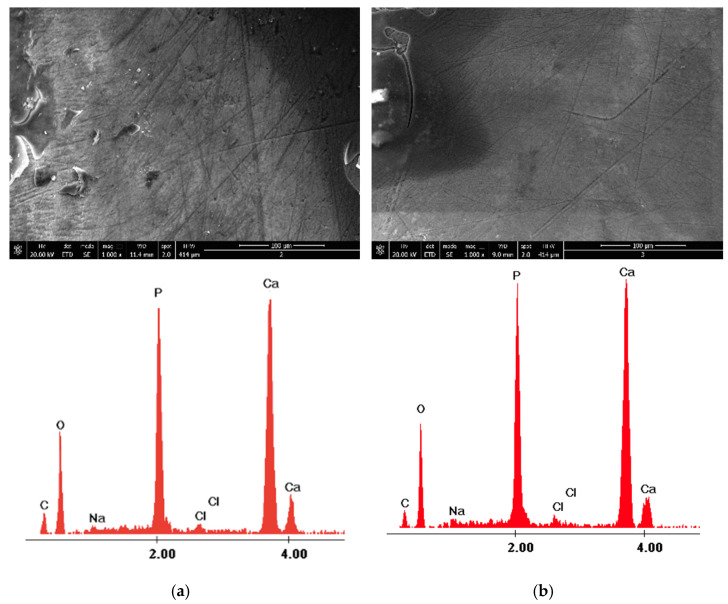
The images recorded by scanning electron microscopy and energy dispersive X-ray analysis for 1064 nm Nd:YAG Laser group: (**a**) before laser irradiation; and (**b**) after laser irradiation.

**Figure 7 medicina-58-00633-f007:**
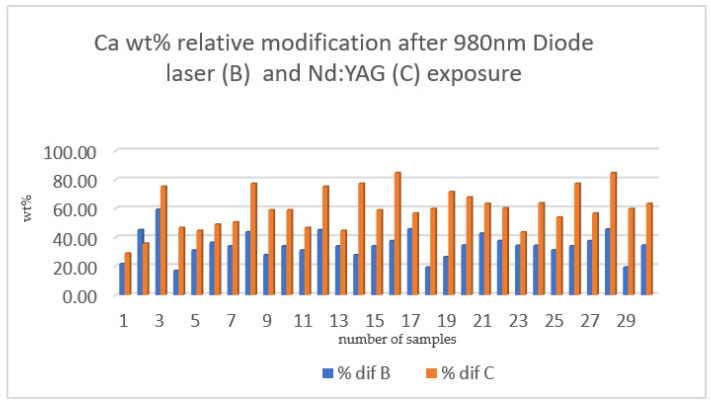
The comparison of calcium weight percentage mean levels recorded after 980 nm diode laser irradiation (% dif B) and after Nd:YAG irradiation (% dif C).

**Figure 8 medicina-58-00633-f008:**
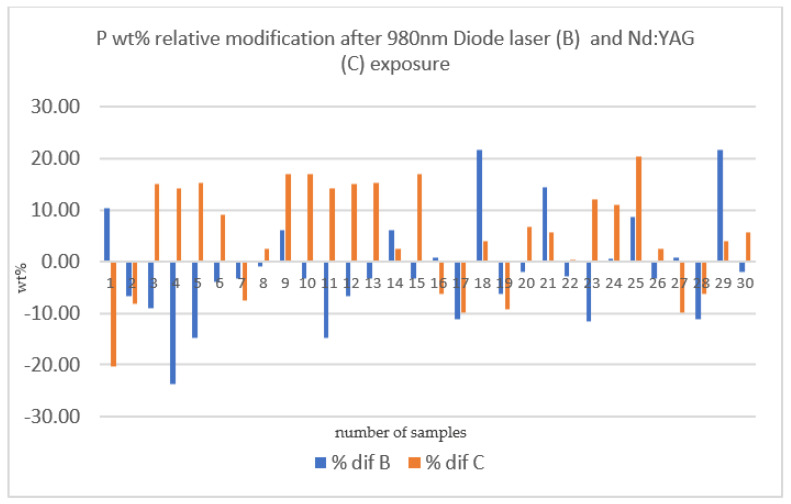
The comparison of phosphorus weight percentage mean levels recorded after 980 nm diode laser irradiation (% dif B) and after Nd:YAG irradiation (% dif C).

**Table 1 medicina-58-00633-t001:** Mean weight percentages of chemical element content in control section and Section B samples before and after 980 nm diode laser irradiation.

	Control Group-Section A	980 nm Diode Laser Group-Section B—before Laser Irradiation	980 nm Diode Laser Group-Section B—after Laser Irradiation
	Mean (SD)	Mean (SD)	Mean (SD)
Ca wt%	21.46 ± 1.72	21.06 ± 3.03	28.24 ± 3.78
P wt%	17.37 ± 2.27	17.20 ± 2.18	16.92 ± 3.02
O wt%	48.40 ± 3.82	49.55 ± 3.85	47.02 ± 5.01
C wt%	11.42 ± 2.07	10.69 ± 2.73	6.72 ± 2.25
Other elements (Na,Cl) wt%	1.27 ± 0.67	1.56 ± 0.82	1.08 ± 0.76

Data are presented as means with standard deviations (SDs).

**Table 2 medicina-58-00633-t002:** The mean weight percentage of chemical element content in Section C samples before and after Nd:YAG irradiation.

	1064 nm Nd:YAG Laser Group-Section C—before Laser Irradiation	1064 nm Nd:YAG Laser Group-Section C—after Laser Irradiation
	Mean(SD)	Mean(SD)
Ca wt%	21.31 ± 1.75	33.88 ± 2.09
P wt%	17.46 ± 2.09	18.28 ± 2.55
O wt%	48.76 ± 2.53	42.89 ± 3.69
C wt%	10.85 ± 1.51	3.69 ± 1.11
Other elements (Na,Cl) wt%	1.63 ± 1.38	1.41 ± 0.80

Data are presented as means with standard deviations (SDs).

**Table 3 medicina-58-00633-t003:** Paired sample statistics of experimental sections before and after laser application, related to the chemical element contents.

		Mean (SD)	N
Pair 1	Ca wt% section B (initial)	21.06 ± 3.03	30
	Ca wt% section B (after laser)	28.24 ± 3.78	30
Pair 2	Ca wt% section C (initial)	21.31 ± 1.75	30
	Ca wt% section C (after laser)	33.88 ± 2.09	30
Pair 3	P wt% section B (initial)	17.20 ± 2.18	30
	P wt% section B (after laser)	16.92 ± 3.02	30
Pair 4	P wt% section C (initial)	17.46 ± 1.97	30
	P wt% section C (after laser)	18.28 ± 2.55	30
Pair 5	O wt% section B (initial)	49.55 ± 3.85	30
	O wt% section B (after laser)	47.02 ± 5.01	30
Pair 6	O wt% section C (initial)	48.76 ± 2.53	30
	O wt% section C (after laser)	42.89 ± 3.69	30
Pair 7	C wt% section B (initial)	10.69 ± 2.73	30
	C wt% section B (after laser)	6.72 ± 2.25	30
Pair 8	C wt% section C (initial)	10.85 ± 1.51	30
	C wt% section C (after laser)	3.69 ± 1.11	30
Pair 9	Other elements wt% section B (initial)	1.56 ± 0.82	30
	Other elements wt% section B (after laser)	1.08 ± 0.76	30
Pair 10	Other elements wt% section C (initial)	1.63 ± 1.38	30
	Other elements wt% section C (after laser)	1.41 ± 0.80	30

N—number of total teeth. SD.—standard deviation.

**Table 4 medicina-58-00633-t004:** Paired sample test results for chemical elements recorded before and after laser application.

		Paired Differences		Significance	
		Mean (SD)	95% CI Lower—Upper	One-Sided *p*	Two-Sided *p*
Pair 1	Ca wt% section B (initial)—Ca wt% section B (after laser)	−7.18 ± 1.88	−7.88; −6.47	<0.001	<0.001
Pair 2	Ca wt% section C (initial)—Ca wt% section C (after laser)	−12.57 ± 2.29	−13.43; −11.71	<0.001	<0.001
Pair 3	P wt% section B (initial)—P wt% section B (after laser)	0.27 ± 1.84	−0.41; 0.96	0.207	0.415
Pair 4	P wt% section C (initial)—P wt% section C (after laser)	−0.82 ± 1.83	−1.50; −0.14	0.010	0.020
Pair 5	O wt% section B (initial)—O wt% section B (after laser)	2.53 ± 3.59	1.18; 3.87	<0.001	<0.001
Pair 6	O wt% section C (initial)—O wt% section C (after laser)	5.86 ± 2.84	4.80; 6.92	<0.001	<0.001
Pair 7	C wt% section B (initial)—C wt% section B (after laser)	3.97 ± 1.84	3.28; 4.66	<0.001	<0.001
Pair 8	C wt% section C (initial)—C wt% section C (after laser)	7.16 ± 1.43	6.62; 7.69	<0.001	<0.001
Pair 9	Others wt% section B (initial)—Others wt% section B (after laser)	0.48 ± 1.17	0.04; 0.92	0.016	0.032
Pair 10	Others wt% section C (initial)—Others wt% section C (after laser)	0.22 ± 1.74	−0.43; 0.87	0.247	0.494

CI—confidence interval of the difference.

**Table 5 medicina-58-00633-t005:** Group statistics for mean differences between wt% of Ca and P, related to both wavelengths used in this study.

	Laser Type	N	Mean (SD)
% DIF OF SECTION B % DIF OF SECTION C related to Calcium	980 nm diode laser	30	34.49 ± 9.10
	Nd:YAG laser	30	59.81 ± 14.06

% DIF OF SECTION B % DIF OF SECTION C related to Phosphorus	980 nm diode laser	30	−1.77 ± 10.16
	Nd:YAG laser	30	4.91 ± 10.59


N—number of samples. Std.—standard.

**Table 6 medicina-58-00633-t006:** The independent samples test for comparison of the 980 nm diode laser and the Nd:YAG laser, with respect to the mean Ca wt% and P wt% before and after irradiation.

t-Test for Equality of Means		Significance		
		df	One-Sided *p*	Two-Sided *p*
% dif B (980 nm diode laser) *% dif C (Nd:YAG)* *Calcium*	Equal variances assumed	58	<0.001	<0.001
	Equal variances not assumed	49.68	<0.001	<0.001
% dif B (980 nm diode laser) *% dif C (Nd:YAG)* *Phosphorus*	Equal variances assumed	58	0.008	0.015
	Equal variances not assumed	57.90	0.008	0.015

dif—difference. N—number of samples. STD.—standard.

## Data Availability

Not applicable.

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
