# Peer review of "Comparative Evaluation of Influence of Nd:YAG Laser (1064 nm) and 980 nm Diode Laser on Enamel around Orthodontic Brackets: An In Vitro Study"

_medicina, 2022, doi:10.3390/medicina58050633_

Round 1

Reviewer 1 Report

The manuscript assessed “In Vitro Comparative Evaluation of Influence of Nd:YAG Laser (1064 nm) and Diode Laser (980 nm) on Enamel Around  Orthodontic Brackets: A Morphological and Elemental Analysis”, this research is under the scope of this Medicina Journal. The topic is not novel and this manuscript does not add to existing knowledge in this field, and there are Major concerns about the present manuscript: 

(Abstract)

  • It's too long. 

(Introduction)

What is the importance of this review study? What is the gap in this field of literature?

    • You do not think this study is included in the others already done? Which results are comparable with other studies? What has this study been new? 
    • The cause and mechanism of white spot lesion formation is not explained clearly. 
    • The authors have put several null hypotheses...however I think the first should be null and the following should be the various alternatives. The alternative hypothesis, however, does not have to be the negation of the null hypothesis.

(M&M)

  • “Intact teeth extracted from young patients with periodontal disease.” -  Young patients would be below 20 years of age … these were adult patients….
  • When mentioning materials or devices: for some of them, you don't mention the manufacturer at all, for some you mention only the manufacturer, for some the manufacturer and city, for some you mention the manufacturer and city/ country.  
  • How was the sample calculated? Did the authors perform a power analysis to evaluate if this sample size was appropriate?
  • Improve the resolution quality of all figures and graphs (and a presentation). The font/language in the figure/caption is different from the text. Please, standardize the size and the font in the figures and charts with the font of the manuscript. 
  • How many operators performed the experiments? Was the operator of the testing SEM and EDX blinded regarding the treatment group? How many times the in vitro (SEM and EDX) study was done? 
  • Image 1 is important to see the Location of the groups, however, the demonisation of Section ABC to identify the groups it´s confusing.  Please standardize the name of the Groups: and my suggestion is control group, Díodo Group, Nd:YAG Group for all the manuscripts.
  • The authors should provide higher magnification images of the original and lased enamel (e.g., at x5000 or x10000) to indicate any impact of lasing on the structure of the hydroxyapatite rods.  SEM images of the enamel are not at appropriate magnifications to appreciate any changes to the structure of the enamel after lasing. 

(Discussion)

  • Please, identified what was the strength(s) and limitations of this study? And also, implications for future perspectives.

(References)

  • Check the reference format in the manuscript and the references. The titles of references have a different format, the title of the article is written in capital letters at the beginning of words, others only in lower case.

Reviewer 2 Report

Poor abstract, revise, add the significant findings in it

Very deficient literature review on characteristics of diode and ND YAG lasers (Materials (Basel). 2021 Dec 6;14(23):7475.; Compend Contin Educ Dent. 2017 Apr;38(eBook 5):e18-e31.;Prog Orthod. 2017 Dec;18(1):15.). also on white spot lesion formation and how to prevent it, Fluoride, nanomaterials, ,,as well as Laser (Bull Tokyo Dent Coll. 2001 May;42(2):79-86.; Am J Orthod Dentofacial Orthop. 2002 Sep;122(3):251-9. ; Dent J (Basel). 2019 Sep 2;7(3):91. ;Acta Odontol Scand. 2014 Aug;72(6):413-7. ).

The study design is not clear and has some issues, needs more explanation of the reproduction of oral conditions, why low vacuum SEM and EDX analysis was used prior to laser treatment as this may change the enamel structure, you should use new fresh enamel.

I don't see remineralisation/ acid-resistance tests on the laser exposed enamel, so how would you know this method is effective to prevent white spot formation, at least suggest future studies to assess these, therefore, the conclusion need modification, you also need better SEM images with higher magnifications images of the original and lased enamel ( x5000 or x10000) to show the impact of laser exposure on the hydroxyapatite rods. ,

Many parts of  the tables are the output of the statistical soft water and not needed, such as DF or standard error, combine mean and Sd and show as mean (SD), same for boundaries of 95 % CI, reduce the figures to 2 decimal, 

Figures in their present form do not add much to the paper.

Discussion, very poor, don't repeat result section, talk about major finding, shortcomings, suggest new studies 

Round 2

Reviewer 1 Report

The authors improved the quality of the manuscript after the reviewer's indications.

Author Response

We would like to thank you very much for your valuable time spent for the review stage and for the valuable suggestions that have helped us to improve the quality of our manuscript. All the aspects you have pointed out have been taken into account and have led to the realization of a valuable work, able to a possible publication.

From all the authors,

Thank you.

Reviewer 2 Report

Thank you for the revision

I have edited the abstract, please add the missing information

Abstract:

(1) Background: The prevention of enamel demineralization continues to represent a challenge in daily dental practice. (talk about P and Ca that you measured, in what way their composition is important in enamel.?

The aim of this study is to evaluate and compare the influence of two laser wavelengths on the surface morphology and mineral components of the enamel, through scanning electron microscopy (SEM) examination and energy- dispersive X-ray spectrometry (EDX). (2) Methods: Thirty permanent human incisors were selected for this study. Metallic brackets (ad the brand name, company, size) were bonded to tooth (explain where you get these teeth). The buccal surface was randomly assigned into three sections: section  A—positive control (no treatment), section B—treated with 980 nm

Gallium–Aluminum–Arsenide diode laser (for how long add power, delivery mechanism, at what distance, what fibre tip diameter?) (KaVo GENTLEray 980 Diode Laser, Biberach, Germany), and section C—treated with Nd:YAG laser (for how long dd power, delivery mechanism, at what distance, what fibre tip diameter?)(LIGHTWALKER AT S, M021-5AF/1 S, Fotona d.o.o, Ljubljana, Slovenia). The elements evaluated in this study were calcium (Ca), phosphate (P), oxygen (O), and carbon (C). (3) Results: Evaluation of the data indicated that both wavelengths produced an increase in Ca wt% (add the changes in 95 % CI for both lasers), while the 980 nm diode laser decreased P wt% and the Nd:YAG laser increased P wt% (where these significant, then add p-value or just trends?).  (4) Conclusions: It can be concluded that the best improvement of enamel chemical composition was obtained with Nd:YAG irradiation. what is the effect of your findings on enamel and decalcification?

 at the end of the introduction, line 218, correct to 'for this  study, the following null hypothesis were considered

add a consort type diagram showing the study samples, interventions, laser characteristics, variables measured, 

results,

table 1, add the diode and Nd:YAG laser with their wavelenthsg to the tale on the top row

table 4, should look like other tables, combine the upper and lower 95%CI boundaries, also remove df and t

table 5, what are B and C, expand/revise?

figures, fr each figure legend, add what is really important for the reader?

reference, remove doi link

introduction and discussion, it is very long, keep the references but shorthen it , revise

Author Response

We would like to thank you very much for your valuable time accorded to the review stage and for the precious suggestions that have helped us to improve the quality of our manuscript. 

From all the authors,

Thank you.

Round 3

Reviewer 2 Report

There are many presentation issues, please spent some time and improve the paper

please resubmit and change the paper presentation so that the reviewer just sees the final version, don't show omissions and deletions, highlight the changes in red

Title, remove 'A Morphological and Elemental Analysis' add 'an invitro study'

Introduction

Remove lines 66-76, as there is no citation, for the next paragraph, expand that populations with poor socioeconomic status present a higher risk of having caries when they have malocclusions (Acta Odontol Scand. 2011;69(1):2-11. )

line 154, please revise and don't invent new terminology!!!, use the common terminology used in other papers, use the following 'for the present  study, the following Null hypothesised were considered:'

Revise figure 2, so many repetitions!!, you need to add the sample number to each group, you just need to mention the variables measured once not for all groups, correct 'Section C - 1064 nm Nd:YAG Laser Group', look at similar papers and see how they produce the consort type diagram

There are many grammar issues, revise, such as line 402'Sundararaj D et al. [363]' which should be'Sundararaj et al. [363]'

line 477, correct to'Umana et al. [34] and Nandkumar and Iyer, [56], or '

F.M. Suhaimi et al. [3229] and Z. Noorsyazwani et al. [451],' or 'El Mansy MM [186]'

I can't locate the references 38 and 39 in the text, better just remove both of them and replace them with an updated one 'Photonics 20229(4),265; https://doi.org/10.3390/photonics9040265)' I think these references belong to line 124-126

Author Response

We would like to thank you very much for your valuable time spent for the review of our article and for the valuable suggestions that have helped us to improve the quality of our manuscript. All the aspects you have pointed out have been taken into account and have led to the realization of a valuable work.

From all the authors,

Thank you.
